# Does bumblebee preference of continuous over interrupted strings in string-pulling tasks indicate means-end comprehension?

**Chao Wen[1,2]\*[†], Yuyi Lu[2,3][†], Cwyn Solvi[4], Shunping Dong[5], Cai Wang[6], Xiujun Wen[6], Haijun Xiao[1], Shikui Dong[1], Junbao Wen[5], Fei Peng[3,4]\*, Lars Chittka[2]\***

[1]School of Grassland Science, Beijing Forestry University, Beijing, China; [2]Biological and Experimental Psychology, School of Biological and Behavioural Sciences, Queen Mary University of London, London, United Kingdom; [3]Department of Psychology, School of Public Health, Southern Medical University, Guangzhou, China; [4]Guangdong-Hong Kong-Macao Greater Bay Area Center for Brain Science and Brain-Inspired Intelligence, Southern Medical University, Guangzhou, China; [5]Beijing Key Laboratory for Forest Pest Control, Beijing Forestry University, Beijing, China; [6]College of Forestry and Landscape Architecture, South China Agricultural University, Guangzhou, China

**\*For correspondence:**
wenchao@bjfu.edu.cn (CW);
fpeng@smu.edu.cn (FP);
l.chittka@qmul.ac.uk (LC)

[†]These authors contributed equally to this work

**Abstract** Bumblebees (*Bombus terrestris*) have been shown to engage in string-pulling behavior to access rewards. The objective of this study was to elucidate whether bumblebees display means-end comprehension in a string-pulling task. We presented bumblebees with two options: one where a string was connected to an artificial flower containing a reward and the other presenting an interrupted string. Bumblebees displayed a consistent preference for pulling connected strings over interrupted ones after training with a stepwise pulling technique. When exposed to novel string colors, bees continued to exhibit a bias towards pulling the connected string. This suggests that bumblebees engage in featural generalization of the visual display of the string connected to the flower in this task. If the view of the string connected to the flower was restricted during the training phase, the proportion of bumblebees choosing the connected strings significantly decreased. Similarly, when the bumblebees were confronted with coiled connected strings during the testing phase, they failed to identify and reject the interrupted strings. This finding underscores the significance of visual consistency in enabling the bumblebees to perform the task successfully. Our results suggest that bumblebees' ability to distinguish between continuous strings and interrupted strings relies on a combination of image matching and associative learning, rather than means-end understanding. These insights contribute to a deeper understanding of the cognitive processes employed by bumblebees when tackling complex spatial tasks.

## eLife assessment

This study provides **valuable** new insights into insect cognition and problem-solving in bumblebees. The authors present **convincing** evidence that bumblebees lack causal understanding in a string-pulling task, and find support for bumblebees instead using image-matching for this task.

## Introduction

String pulling is one of the most extensively used approaches in comparative psychology to evaluate the understanding of causal relationships (*Jacobs and Osvath, 2015*), with most research focused on mammals and birds, where a food item is visible to the animal but accessible only by pulling on a string attached to the reward (*Taylor et al., 2010*; *Range et al., 2012*; *Jacobs and Osvath, 2015*; *Wakonig et al., 2021*). A fundamental challenge in animal cognition research revolves around unraveling the strategies that animals employ when confronted with specific tasks (*de Waal and Ferrari, 2010*; *Shettleworth, 2010*; *Chittka et al., 2012*). The complexity of the string-pulling paradigm can be altered by manipulating the number and mutual positions of the strings and reward, allowing the investigation of different aspects of cognition (*Jacobs and Osvath, 2015*; *Wang et al., 2019*). Multiple mechanisms can be involved in the string-pulling task, including the proximity principle, perceptual feedback and means-end understanding (*Taylor et al., 2012*; *Wasserman et al., 2013*; *Jacobs and Osvath, 2015*; *Wang et al., 2021*). The principle of proximity refers to animals preferring to pull the reward that is closest to them (*Jacobs and Osvath, 2015*). *Taylor et al., 2012* proposed that the success of New Caledonian crows in string-pulling tasks is based on a perceptual-motor feedback loop, where the reward gradually moves closer to the animal as they pull the strings. If the visual signal of the reward approaching is restricted, crows with no prior string-pulling experience are unable to solve the broken string task (*Taylor et al., 2012*).

Means-end understanding is expressed as goal-directed behavior, which involves the deliberate and planned execution of a sequence of steps to achieve a goal (*Jacobs and Osvath, 2015*; *Torres Ortiz et al., 2019*). String-pulling studies have directly tested means-end comprehension in various species (*Riemer et al., 2014*; *Jacobs and Osvath, 2015*). In these studies, organisms are presented with two or more strings, where one is connected to a reward and the other one is interrupted; pulling the connected string indicates that animals comprehend that a continuous string is a means to the end of obtaining the reward (*Piaget, 1953*; *Wasserman et al., 2013*; *Jacobs and Osvath, 2015*; *Hofmann et al., 2016*; *Wang et al., 2019*). Most animals fail in such string-pulling tasks when they have to spontaneously solve them, but they can be trained to recognize that an interrupted string is useless through trial-and-error learning (*Mayer et al., 2014*; *Torres Ortiz et al., 2019*; *Wang et al., 2019*).

To our knowledge, bumblebees are the only invertebrates that have been trained to learn to pull a string to obtain a reward (*Alem et al., 2016*; *Wen et al., 2024*; *Zhou et al., 2024*). However, the performance of bumblebees in these studies could be explained by associative learning (*Alem et al., 2016*). It remains unknown whether the bumblebees understand the connectivity of the string. We aim to explore the question of means-end comprehension in bumblebees when tackling string-pulling tasks. We conducted nine horizontal string-pulling experiments. These experiments were designed to manipulate factors such as string color and spatial arrangement during both the training and testing phases. Firstly, bumblebees without string-pulling experience were tested to discriminate between strings connected to a target containing the reward and disconnected strings. In another set of experiments, we examined whether bumblebees with string-pulling experience would discriminate between connected and disconnected strings. Furthermore, we changed the color of strings in training to determine whether bees generalize features learned to solve tasks with different colored strings. In two other experiments, black tape was used to cover the strings, preventing the bees from seeing the string connected to the flower from above the table during the training phase. In one of these experiments, a green table was placed behind where the bee pulled to prevent the bee from seeing the string connected to the flower even after having reached the reward during the training phase. Finally, to further verify whether bumblebees choose strings through image learning, the straight strings were changed to coiled so that the image of the string was distinct from that in training.

## Results

In Experiment 1, we evaluated whether uninformed bumblebees with no string-pulling experience could discriminate between connected and disconnected strings. Bees were trained to retrieve a yellow flower (without a string) containing sugar water from under a transparent table (*Figure 1*; *Video 1*). In the test, bees had two different flowers to choose from, one with a connected string and the other with a disconnected string (*Figure 2*). The strings and flowers used were discarded after

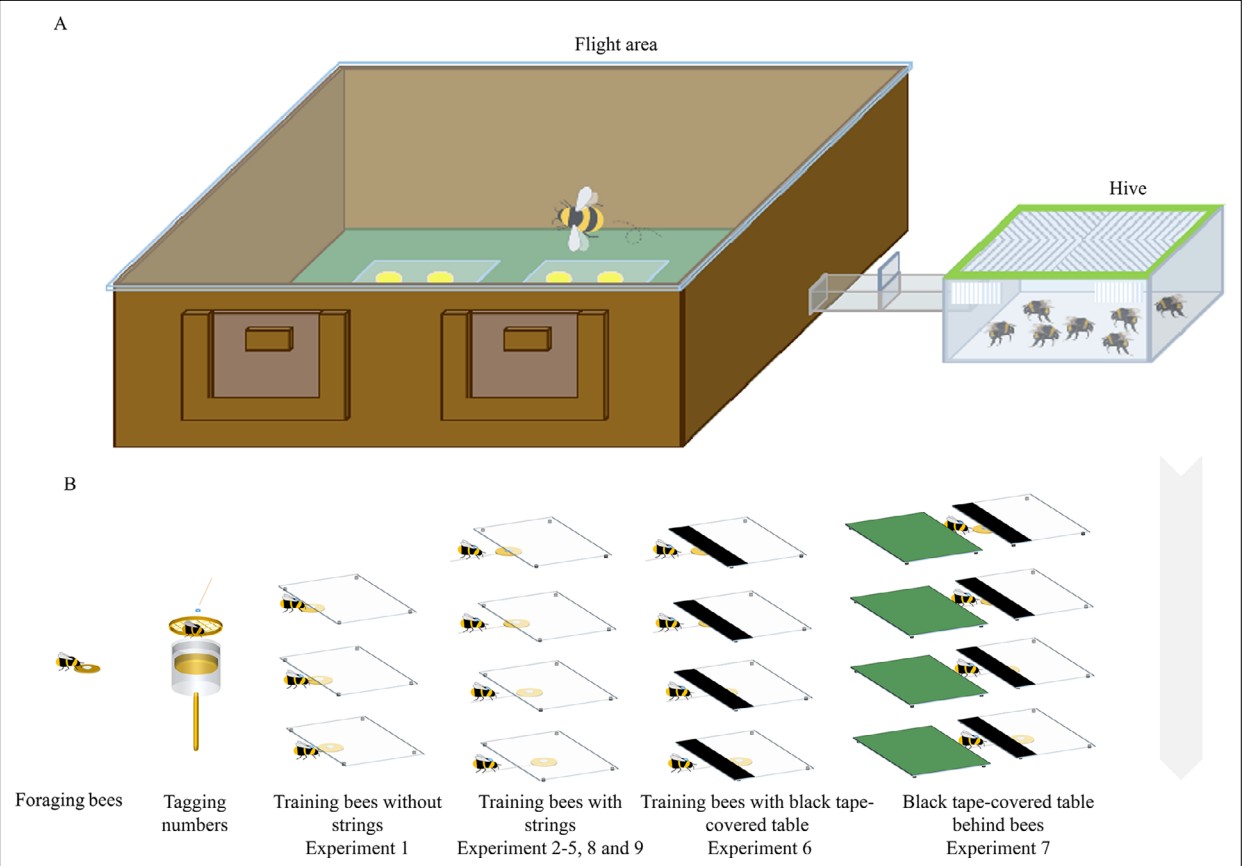

**Figure 1.** Experimental apparatus and summary of training in different experiments. (**A**) The experimental setup consisted of a flight arena connected to a hive via a Perspex corridor. (**B**) Foraging bumblebees were number-tagged, and the marked bees were trained in a stepwise manner.

each test to prevent the use of chemosensory cues. We found that 10 of 21 bees (48%, $\chi^2$=0.05, p=0.83) chose the connected string as their first choice in the test (*Table 1*). Over the entire duration of the test, bees showed no preference for either string during the test [n=21, generalized linear mixed model (GLMM): 95% CI=−0.09 (-0.48–0.29), Z = −0.48, p=0.63; *Figure 3*], suggesting that there is no spontaneous comprehension of the significance of the gap, and that any preference for connected strings would have to be acquired through training.

We next examined whether bumblebees with string-pulling experience would discriminate between connected and disconnected strings. In Experiment 2, bees were first trained to retrieve yellow flowers from under a transparent table by pulling an attached white string (*Video 2*). After training, bees were presented with two different flowers: one was connected to a string and the other had a disconnected string. We found that 13 of 18 bumblebees (72%, $\chi^2$=3.56, p=0.06) chose the connected string as their first choice in the test (*Table 1*). Over the entire duration of the test, bees selected the connected strings (76 ± 4%) significantly more than the disconnected strings [n=18, GLMM: 95% CI=1.06 (0.78–1.33), Z=7.56, p<0.001; *Figure 3*, *Video 3*]. In addition, bees spent much more time attempting to pull the connected strings (94.67±13.19 s) than the disconnected strings (13.61±4.59 s) [n=18, GLMM: 95% CI=0.07 (0.02–0.11), t=2.90, p<0.01; *Figure 4*].

In Experiment 3, we trained another group of bees with the same procedure as in Experiment 2, but the disconnected string pointed to the flower in the test (there was no lateral displacement between the disconnected string and flower in the test; *Figure 2*). Again, we found that most bees (17/18, 94%, $\chi^2$=14.22, p<0.001) chose the connected string as their first choice (*Table 1*). Over the course of the entire test, the percentage of bees pulling connected strings (79 ± 4%) was significantly above chance level [n=18, GLMM: 95% CI=1.56 (0.72–2.40), Z=5.90, p<0.001; *Figure 3*; *Video 4*], and bees spent longer times manipulating connected strings (79.59±10.15 s) [GLMM: 95% CI=0.09 (0.03–0.14), t=3.23, p<0.01; *Figure 4*].

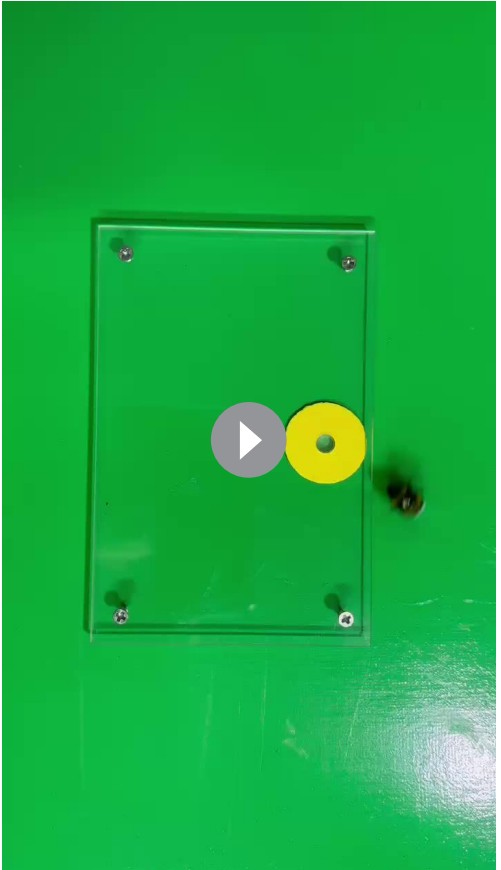

**Video 1.** Training the bumblebees without string. Bees were trained to retrieve a yellow flower (without a string) containing sugar water from under a transparent table.

https://elifesciences.org/articles/97018/figures#video1

To explore whether bees simply memorized the visual display of the 'lollipop shape' of a string connected to a flower during training with a given color combination, we then asked whether bees generalize from the string color used during training. Bumblebees were trained with green strings (*Video 5*) or blue strings (*Video 6*) connected to flowers. If the bees had the ability to generalize the function of strings, this color change should not affect their ability to discriminate connected and disconnected strings. Overall, changing the color of the string in training reduced the accuracy of the choice, 7/10 bees (70%, $\chi^2$=1.60, p=0.21) selected the connected string in the first choice (*Table 1*). But over the entire test, the bees still maintained the basic discrimination at well above chance level when trained with green strings (61 ± 5%) [n=10, generalized linear model (GLM): 95% CI=1.13 (0.71–1.54), $Z$=5.27, p<0.001; *Figure 3*]. In addition, the duration of pulling connected strings (47.79±9.91 s) was significantly longer than disconnected ones (24.78±6.05 s) [n=10, GLMM: 95% CI=0.10 (0.01–0.19), $t$=2.26, p<0.05; *Figure 4*]. In this sense, bees might possess the ability to generalize string color from the originally learned stimulus. A similar result was found in bees trained with blue strings. We found that 14 of 16 bees (87%, $\chi^2$=9.00, p<0.05) selected the connected string in the first choice (*Table 1*). The percentage of bees pulling connected strings was significantly higher than chance level [n=16, GLMM: 95% CI=0.60 (0.30–0.90), $Z$=3.90, p<0.001; *Figure 3*], and the duration data also indicate that the bees prefer pulling connected strings [n=16, linear mixed model (LMM): 95% CI=−66.91 (-80.86 to -52.96), $t$ = −9.68, p<0.001; *Figure 4*].

The question of whether animals rely on perceptual feedback during string pulling has been tested with occluders between the string and the reward in a variety of other species (*Taylor et al., 2010*; *Gaycken et al., 2019*; *Chaves Molina et al., 2019*; *Wakonig et al., 2021*). Bumblebees were trained to feed on yellow artificial flowers, and then trained with transparent tables covered by black tape through a four-step process (*Video 7*). The aim was to prevent the bees from seeing the string connected to the flower during training. Note, however, the bees were able to see this 'lollipop shape' (string connected to the flower) after they pulled the strings out from the table or during the initial step of the training (*Figure 1*). Ten out of fifteen bees (67%, $\chi^2$=1.67, p=0.20) pulled the connected string in their first choice (*Table 1*). However, over the full duration of the test, the percentage of the bees pulling connected strings (82 ± 3%) was significantly higher than chance level [n=15, GLMM: 95% CI=1.31 (1.00–1.62), $Z$=8.23, p<0.001; *Figure 3*], and the duration data also indicate that the bees preferred pulling connected strings [n=15, GLMM: 95% CI=0.10 (0.02–0.17), $t$=2.61, p<0.01; *Figure 4*]. To help ensure that the bees could not see the 'lollipop shape' during the training phase, we placed a green table behind the bee (*Figure 2*). In this way, the 'lollipop shape' was not directly presented during the initial step of training, nor was it visible after the bee had reached the reward because the string was then covered by the back green table (*Video 8*). We found that the initial choices of bees in this test were at chance level (5 out of 10 bees chose the connected string). Over the full test, the percentage of pulling connected strings was significantly lower than chance level

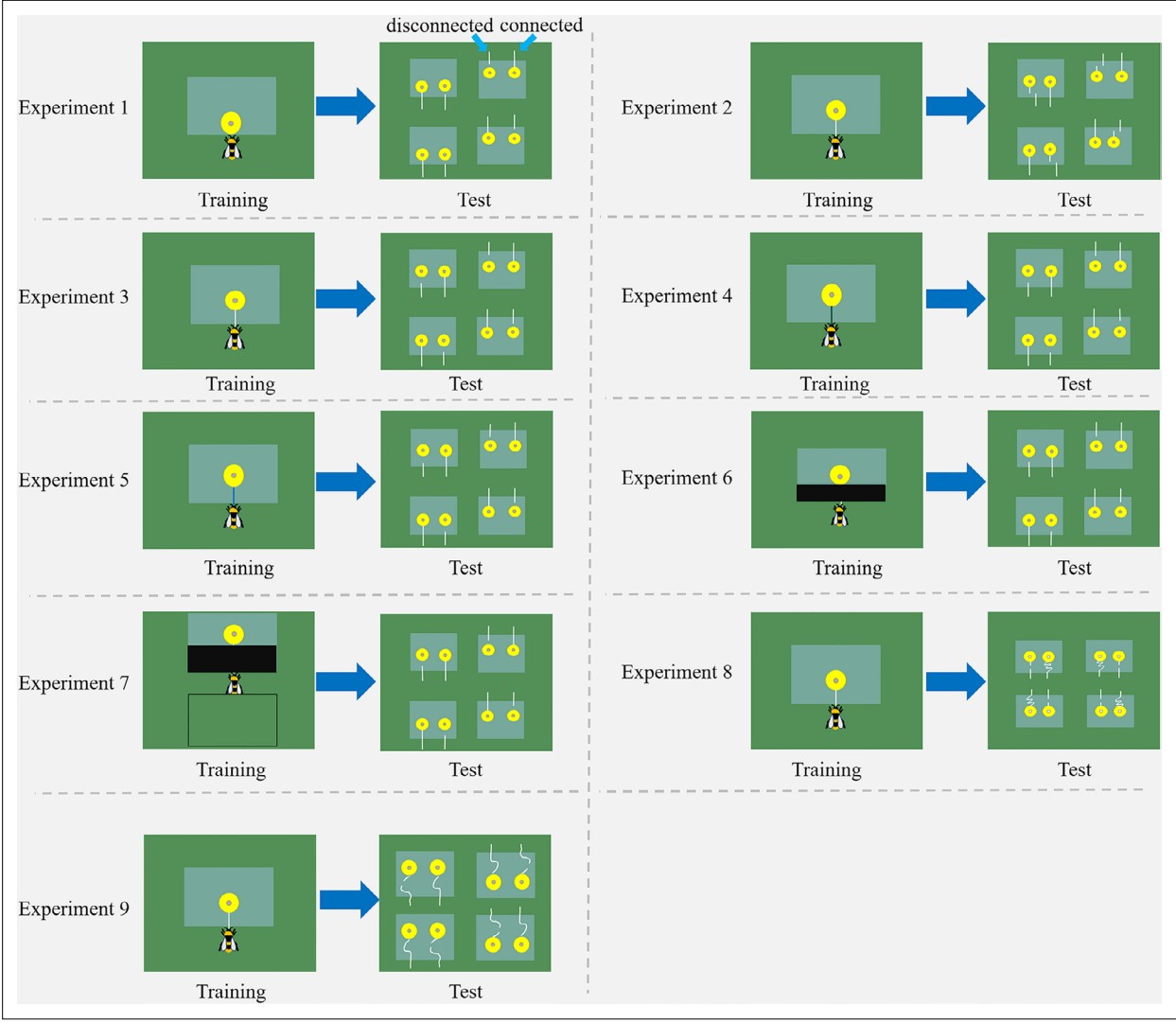

**Figure 2.** Schemes of nine string-pulling experiments. Bumblebees were trained and tested in different situations. Two flowers were placed under each transparent table in the test, with one string connected to a flower and another was detached. For further details and descriptions of each experiment, see the text.

[n=10, GLMM: 95% CI=−0.77 (-1.33 to -0.20), Z = −2.61, p<0.01; *Figure 3*]. But duration data indicated that the bees showed no preference for pulling the connected strings or disconnected strings [n=10, GLMM: 95% CI=−0.09 (-0.38–0.19), t = –0.64, p=0.53; *Figure 4*].

So far, our results show that bumblebees could have been using image matching to discriminate connected from disconnected strings in the test. We therefore designed further experiments based on *Taylor et al., 2012* to test this hypothesis. Bumblebees were first trained to feed on yellow artificial flowers, and then trained with the same procedure as Experiment 2, but the connected strings were coiled in the test. Bees failed to choose the connected strings when strings were coiled. We observed that 11/20 bees (55%, $\chi^2$=0.20, p=0.65) pulled the connected string in their first choice (*Table 1*). Over the full duration of the test, no difference in the percentage of pulling connected strings compared with chance level [n=20, GLMM: 95% CI=0.17 (-0.26–0.59), Z=0.70, p=0.48; *Figure 3*]. There was also no significant difference between the duration of bees pulling the connected strings and disconnected strings [n=20, LMM: 95% CI=−5.77 (-24.04–12.50), t = –0.63, p=0.53; *Figure 4*], suggesting that bumblebees did not recognize the continuity of coiled strings. Another group of bees was trained with straight strings, while both connected and disconnected strings were coiled in the test. Ten of 19 bees (53%, $\chi^2$=0.05, p=0.82) chose the connected string as their first choice in the test (*Table 1*). Similarly, no differences were found in percentage [n=19, GLMM: 95% CI=−0.03 (-0.54–0.48), Z =

**Table 1.** The number of total choices and bees that chose connected strings at first choice.

| Experiment | 1 | 2 | 3 | 4 | 5 | 6 | 7 | 8 | 9 |
|---|---|---|---|---|---|---|---|---|---|
| Number of choices (mean ±SE) | 5.10±0.54 | 15.00±1.53 | 15.56±2.16 | 12.10±2.19 | 11.63±1.66 | 15.87±2.05 | 5.60±1.55 | 20.60±1.80 | 14.37±2.82 |
| Number of bees choosing connected strings at first choice/N | 10/21 (N.S.) | 13/18 (N.S.) | 17/18 (***) | 7/10 (N.S.) | 14/16 (*) | 10/15 (N.S.) | 5/10 (N.S.) | 11/20 (N.S.) | 10/19 (N.S.) |
| Chi-square result | $\chi 2=0.05$ $P=0.83$ | $\chi 2=3.56$ $P=0.06$ | $\chi 2=14.22$ $P<0.001$ | $\chi 2=1.60$ $P=0.21$ | $\chi 2=9.00$ $P<0.05$ | $\chi 2=1.67$ $P=0.20$ | $\chi 2=0.00$ $P=1.00$ | $\chi 2=0.20$ $P=0.65$ | $\chi 2=0.05$ $P=0.82$ |

The numbers before the slash are the numbers of bumblebees pulling connected strings on the first choice, and the numbers after the slash are the total number of tested bumblebees. N is the total number of the bees. *** p<0.001; * p<0.05; N.S. p>0.05.

–0.29, p=0.77; *Figure 3*] and duration [n=19, GLMM: 95% CI=0.005 (-0.02–0.03), *t*=0.22, p=0.83; *Figure 4*] of pulling each string.

Latency to the first choice was defined as the latency to initiate pulling the string after the bee entered the set-up. The latency to the first choice was measured to assess if the bumblebees were

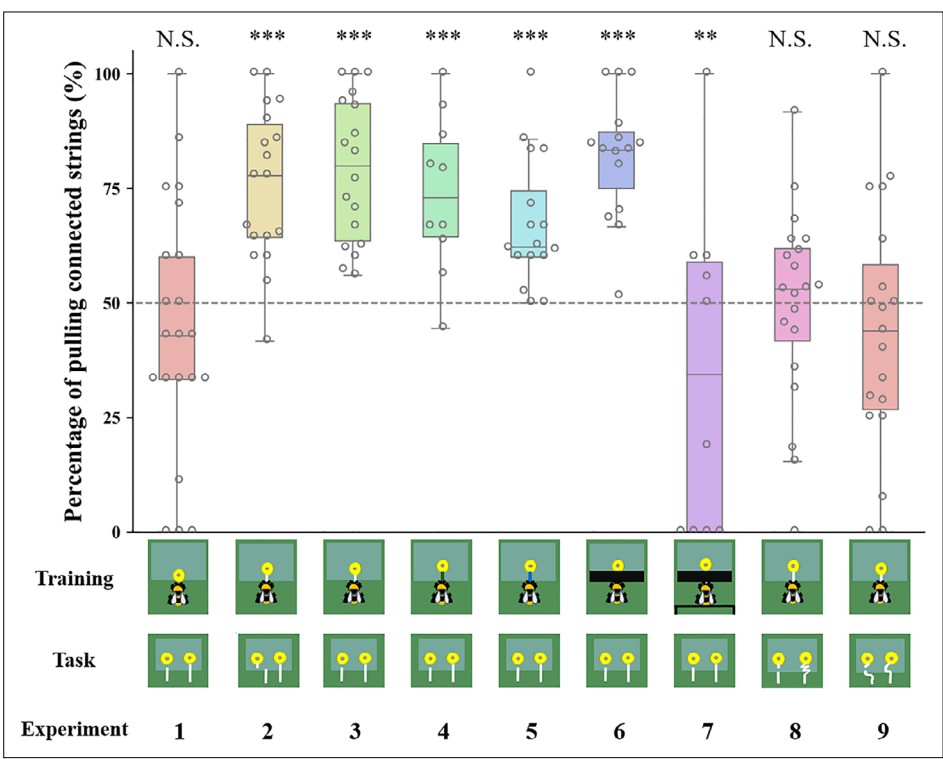

**Figure 3.** Bumblebee preferences for continuous strings over the entire duration of the test in different string-pulling experiments. Percentage of pulling connected strings compared with chance level (50%). Data are presented as mean ± SE. Boxes show the 25th percentile, 50th percentile (median), 75th percentile. The dashed line represents chance level (50%) and the circles indicate individual bees' data points. *** p<0.001; ** p<0.01; N.S. p>0.05.

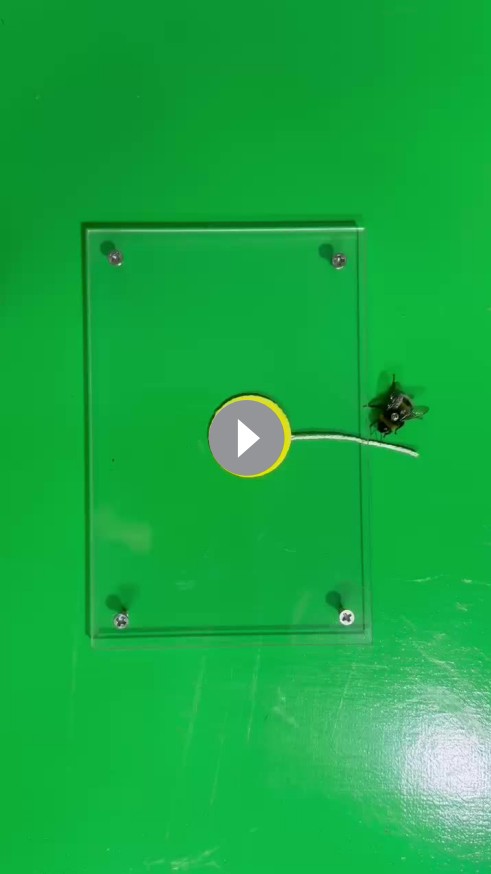

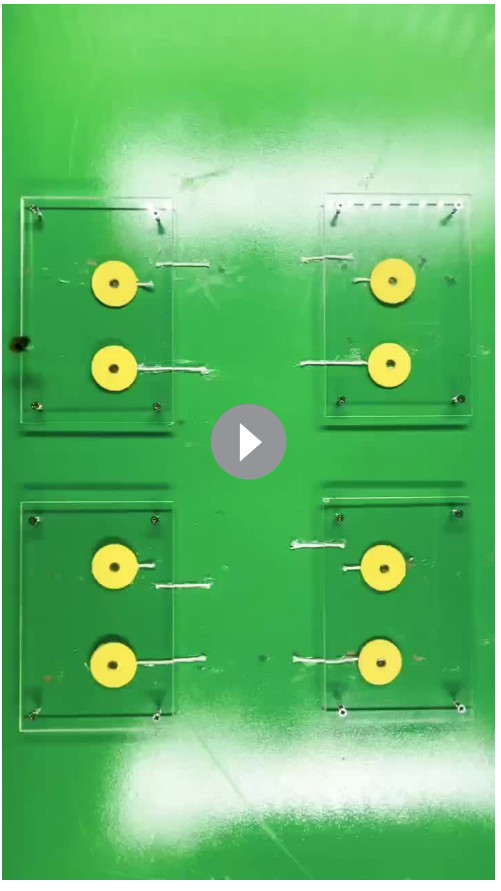

**Video 2.** Training the bumblebees with a white string. Bees were trained to retrieve yellow flowers from under a transparent table by pulling an attached white string.
https://elifesciences.org/articles/97018/figures#video2

**Video 3.** String-pulling test of bumblebee in Experiment 2; the footage shows a bee attempting to pull the connected strings rather than the disconnected strings.
https://elifesciences.org/articles/97018/figures#video3

familiarizing themselves with the testing pattern.

A shorter latency might indicate that the bumblebees were more familiar with the patterns. The latency of the bees that were trained with blue strings (684.33±105.45 s) was substantially longer than that of bees which were trained with white strings and tested with straight strings (*Figure 5*). A long latency time was observed in Experiment 7 (557.26±104.09 s), in which the bees were trained with black tape covering the table, and a green table placed behind the bees (*Figure 5*).

## Discussion

Our results show that: (i) bumblebees require experience with string pulling to distinguish between connected and disconnected strings; (ii) bumblebees are able to generalize features learned during string-pulling training to solve a task with different colored strings; (iii) bumblebees solve string-pulling tasks through image matching.

The results suggest that bumblebees require experience to recognize interrupted strings and acquire a preference for connected ones. This corroborates previous findings that most bees failed to solve single string-pulling tasks without previous training (*Alem et al., 2016*). Some animals, including dogs (*Osthaus et al., 2005*), cats (*Whitt et al., 2009*), western scrub-jays (*Hofmann et al., 2016*) and azure-winged magpies (*Wang et al., 2019*) fail in such spontaneous tasks. It is worth noting that some crows and parrots known for complex cognition perform poorly on the broken string task without perceptual feedback or learning. For example, New Caledonian crows use perceptual feedback strategies to solve the broken string-pulling task, and no individual showed a significant preference for the connected string when perceptual feedback was restricted (*Taylor et al., 2012*). Some

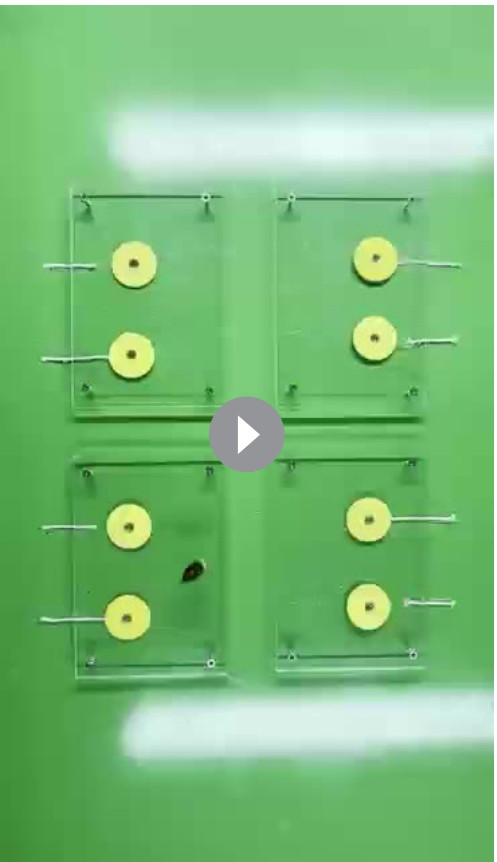

**Video 4.** String-pulling test of bumblebee in Experiment 3; the footage shows a bee attempting to pull the connected strings rather than the disconnected strings.

https://elifesciences.org/articles/97018/figures#video4

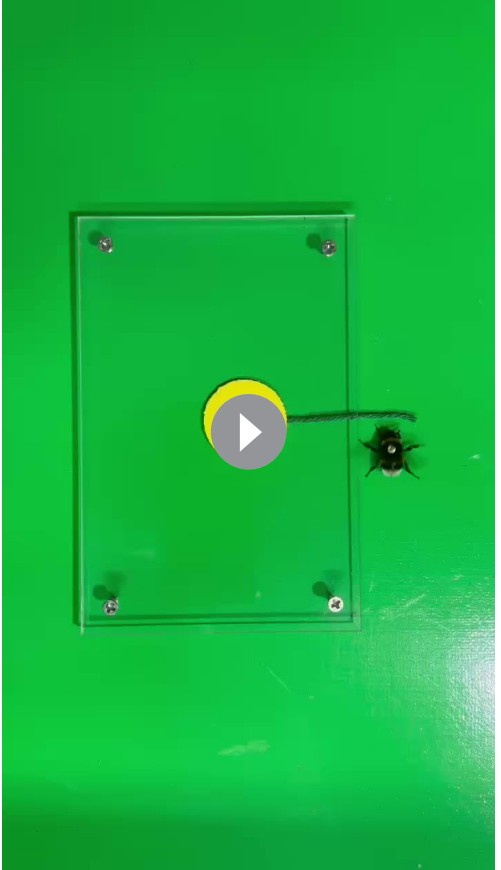

**Video 5.** Training bumblebees with a green string. Bees were trained to retrieve yellow flowers from under a transparent table by pulling an attached green string, to test whether bees could generalize from the string color used during training.

https://elifesciences.org/articles/97018/figures#video5

Australian magpies and African grey parrots can solve the broken string task, but they require a high number of trials, indicating that learning plays a crucial role in solving this task (*Chaves Molina et al., 2019*; *Johnsson et al., 2023*).

Our findings suggest that bumblebees with experience of string pulling prefer the connected strings, but they failed to identify the interrupted strings when the string was coiled in the test. Bees acquire their preferences for flowers with connected strings at least in part by learning the visual appearance of the 'lollipop shape' present during training. Trained bees may have memorized this image as a predictor of reward and applied it to solve novel string-pulling tasks. This makes sense because the bees could see the 'lollipop shape' once they pulled it out from the table in Experiment 6. Another possibility is that bumblebees might remember the image of the 'lollipop shape' from the initial training step, because the shape was directly presented to the bees. However, when a green table was placed behind the string to obscure the 'lollipop' structure during the training, the bees could not see the 'lollipop' during the initial training stage or after pulling the string from under the table. In this situation, the bees were unable to identify the connected string, further supporting the notion that bumblebees chose the connected string based on image matching. Bumblebees exhibited longer delay times to the first choice in Experiments 5 and 7 (*Figure 5*). The reason might be attributable to bees' search time for the familiar image or neophobic response (*Muller et al., 2010*), because the strings during training were changed in the test, which the bees had not encountered before.

Bees often need to match memorized images of flowers to currently visible flowers while foraging in the wild (*Chittka et al., 1999*; *Giurfa, 2003*). Different flower species offer varying profitability in

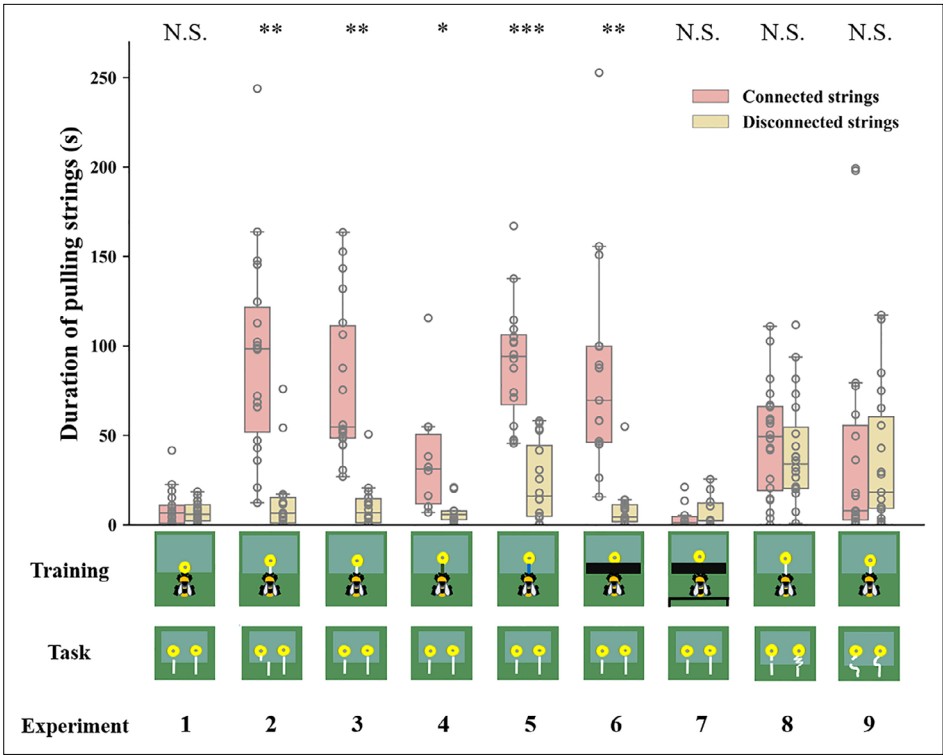

**Figure 4.** Duration of bumblebees attempting to pull the strings over the entire duration of the test across experiments. Median, interquartile range and range are given. Boxes show the 25th percentile, 50th percentile (median), 75th percentile. Circles indicate individual bees' data points. *** p<0.001; ** p<0.01; * p<0.05; N.S. p>0.05.

terms of nectar and pollen to bumblebees; they need to make careful choices and learn to use floral cues to predict rewards (*Chittka, 2017*). Bumblebees can easily learn visual patterns and shapes of flowers (*Meyer-Rochow, 2019*) they can detect stimuli and discriminate between differently coloured stimuli when presented as briefly as 25ms (*Nityananda et al., 2014*). In contrast, causal reasoning involves understanding and responding to causal relationships. Bumblebees might favor, or be limited to, a visual approach, likely due to the efficiency and simplicity of processing visual cues to solve the string-pulling task.

Rather than relying on means-end comprehension, most animals likely use simpler associative strategies to solve string-pulling tasks, as observed in the bumblebees in the present study (*Jacobs and Osvath, 2015*; *Wakonig et al., 2021*). Empirical evidence from several vertebrates has shown that success in object use does not necessarily imply causal understanding; rather, it involves abstracting simple rules based on observable features of the physical task at hand (*Seed et al., 2006*; *Schuck-Paim et al., 2009*; *Herrmann et al., 2008*; *Gagne et al., 2012*). In several vertebrate string-pulling studies, animals relied on a 'proximity rule' in most cases, choosing to pull the strings closest to the reward, regardless of their connectivity (*Whitt et al., 2009*; *Wang et al., 2021*).

In conclusion, even though bumblebees may not understand the causality of string-pulling, they can match the image of the strings connected to flowers, and rely on associative mechanisms to remember the previously visited stimuli. Bumblebees, whether with or without string-pulling experience, do not appear to understand the value of strings to target objects. This negative result does not necessarily mean that bumblebees are entirely unable to comprehend the link between a means and an end, but our results suggest that for the paradigms tested here, such comprehension is not required.

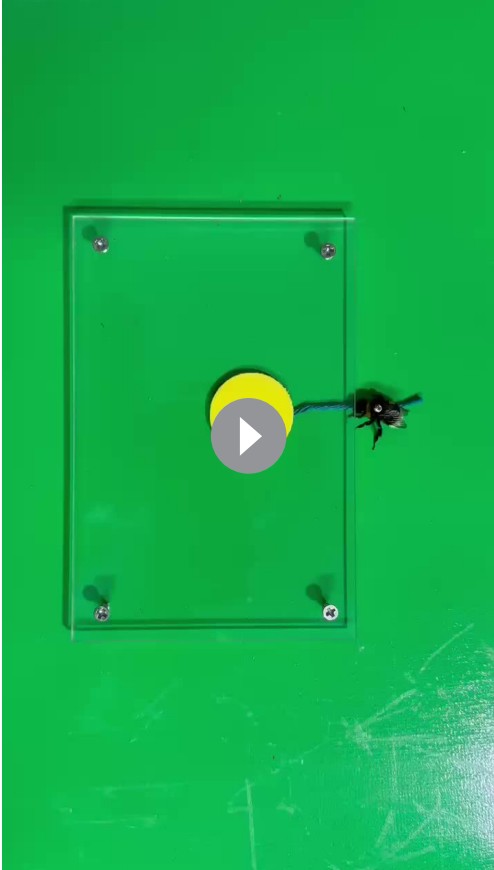

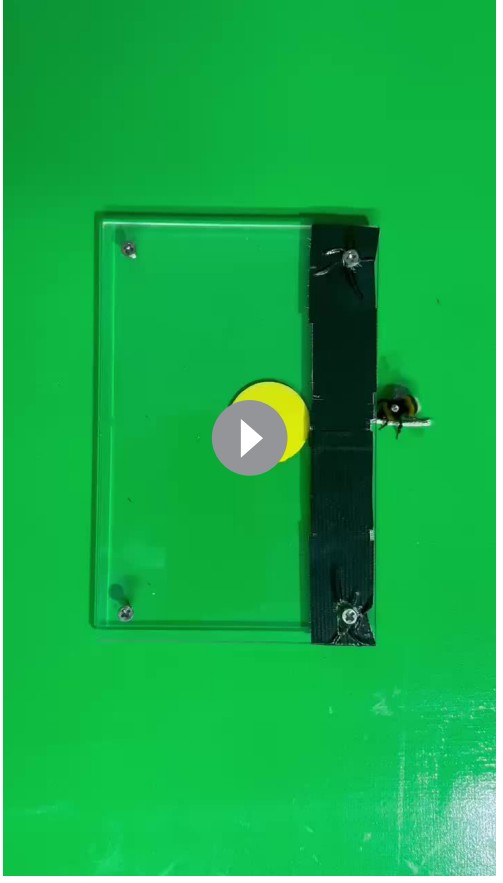

**Video 6.** Training bumblebees with a blue string. Bees were trained to retrieve yellow flowers from under a transparent table by pulling an attached blue string, to test whether bees could generalize from the string color used during training.

https://elifesciences.org/articles/97018/figures#video6

**Video 7.** Training bumblebees with a black tape-covered table. Bees were trained to retrieve yellow flowers from under a transparent table covered by black tape, hypothesizing that bees were not able see the movement of the string above the table.

https://elifesciences.org/articles/97018/figures#video7

## Materials and methods

### Animals and experimental arena

Nineteen colonies of bumblebees (*Bombus terrestris*) each containing a queen were obtained from commercially available stocks provided by a distributor in the United Kingdom (Biobest, Belgium N.V.) or China (Biobest, Belgium N.V.; Biobest Shouguang Biotechnology Co., Ltd). Bees were housed in a plastic nest box (29×22.5 × 13.3 cm [L×W × H]) that was connected to a flight arena (100×75 × 30 cm [L×W × H]) by an acrylic corridor (25×3.5 × 3.5 cm [L×W × H]). The flight arena was covered with an acrylic lid. Three sliding doors were placed along the corridor, allowing the experimenter to control bees' access to the arena (*Figure 1A*). The floor of the flight arena was painted green, which provided a smooth surface and high-contrast pattern visual panorama between the strings and background for the bees (*Spaethe et al., 2001*). Outside of experiments, the colonies were provided with 20% (w/w; weight-to-weight) sucrose solution from a gravity feeder placed in the center of the arena, and with ~5 g commercially obtained pollen (Koppert B.V., The Netherlands; Changge Yafei Beekeeping Professional Cooperative, China) every other day. All the training programs and tests were conducted in the flight arena between 9 am and 7 pm under light (12 : 12 hr) at room temperature (23±4°C). Illumination was provided by fluorescent lighting (Osram Sylvania, Wilmington, NC, U.S.A.; YZ36RR, 36 W, T8/765, FSL, China) fitted with high-frequency ballasts (HFB 236 TLD, Philips, Amsterdam, The Netherlands; T8, YZ-36 W, FSL, China) to generate lighting above the bee flicker fusion frequency (*Skorupski and Chittka, 2010*).

Before each experiment, bees were first pretrained to find sucrose solution (50%, w/w) in yellow artificial flowers (3 cm diameter yellow discs with an inverted Eppendorf cap at the center; henceforth 'flowers'), which were randomly located in the arena with sucrose solution (50%, w/w) in the Eppendorf cap. Bees that seemed to forage with regularity were number-tagged for individual identification (*Figure 1B*). In detail, one forager bee was transferred to a cylindrical cage (diameter = 3.8 cm, length = 7.7 cm) with a sponge plunger, and a numbered tag (Bienen-Voigt & Warnholz GmbH & Co. KG, Germany) was glued to bee's thorax for individual recognition (*Figure 1B*).

## General methods

For each experiment, bumblebees were trained to retrieve a flower with an inverted Eppendorf cap at the center, containing 25 microliters of 50% sucrose solution, from underneath a transparent acrylic table [0.6 cm above the ground, 15×10 × 0.4 cm (L×W × H); henceforth 'table']. For Experiment 1, bees were trained in a stepwise manner – Step 1, 50% of the flower was covered by the transparent acrylic table, Step 2, 75% of the flower was covered; Step 3, 100% of the flower was covered (*Figure 1B*). For Experiment 2–8, the first three steps were similar to Experiment 1, but the flowers were connected to strings (length = 4.5 cm), and bees were trained with the fourth step: 2 cm strings were attached to the flower and accessible from outside the table. The bees received rewards five times in each of the steps, except for the last step. The training phase was completed when a bee pulled the strings and drank from the flowers twenty times after the first occurrence of string pulling during the last step. For each test, bees were individually tested in the arena and presented with four transparent tables. Two options were placed under each table, parallel to each other, and perpendicular to the long side of the table. To avoid developing a side bias, the position and direction of the strings varied randomly from left to right for each table. During tests, both strings were glued to the floor of the arena to prevent the air flow generated by flying bumblebees' wings from changing the position of the string. Different groups of bees were tested for Experiments 1–9, and each forager bee was used only once, and the tested bees were removed from the nest and then placed in the freezer to be euthanized. All the experiments were videotaped with an iPhone 12 (Apple, Cupertino, CA, USA) placed above the arena. The choice and duration of the bees pulling the connected or disconnected strings were recorded. A choice was recorded when a bee used her legs or mandibles to pull the connected or disconnected strings. The test was terminated when the bee stopped engaging with the tables, flowers and strings for more than one minute, and a testing session lasted a maximum of 30 min.

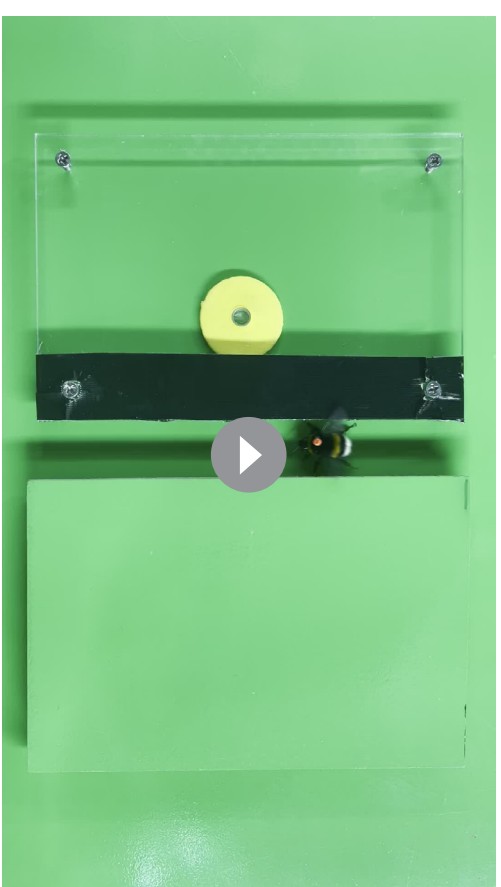

**Video 8.** Training the bee with a black tape-covered table, and a green table behind the bee, hypothesizing that bees were not able to see the image during the first step of training, and the string disappeared from the bees' view when the flower was pulled out from under the table.

https://elifesciences.org/articles/97018/figures#video8

## Experiment 1: Do bumblebees without string-pulling experience discriminate between connected and disconnected strings?

Bumblebees (n=21) were trained with artificial flowers under a transparent table. Initially, half of the flower was placed under the table, and the central Eppendorf cap (containing the sucrose solution reward) at the edge of the table so that the bees could access sugar water directly without

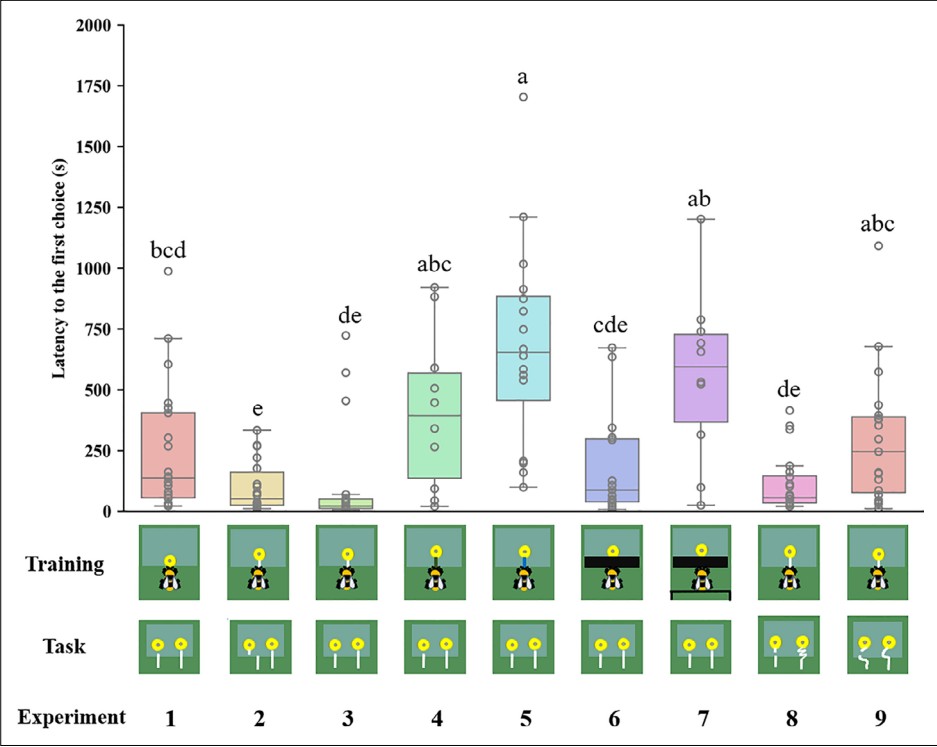

**Figure 5.** Latency to the first choice in different experiments. Median, interquartile range and range are given. Boxes show the 25th percentile, 50th percentile (median), 75th percentile. Circles indicate individual bees' data points. Different letters indicate significant differences (p<0.05).

moving the flower (*Figure 1B*). In the final step, the edge of the flower was aligned with the table, and bees had to pull the edge of the flower to obtain the reward (*Figure 1B*). In the test, four transparent tables were placed on the arena floor, and two artificial flowers were placed 3 cm apart under each table, with one connected to a string (length = 2.5 cm, including 1 cm accessible from outside the table, diameter = 0.3 cm), while another flower was presented with a 1.5 cm string with a cm gap between the string segment and the flower (*Figure 2*). If the bees prefer to pull the connected strings, this would indicate that bees naturally recognize the connectivity of strings in this task.

## Experiments 2-3: Do bumblebees with string-pulling experience discriminate between connected and disconnected strings?

In Experiment 2, bumblebees (n=18) were trained to pull a white string (length = 4.5 cm, diameter = 0.3 cm) attached to the yellow artificial flower that was placed under a transparent table, following the stepwise string-pulling training protocol used by *Alem et al., 2016* with some modifications. Briefly, selected bees were trained to pull the string when the flowers were gradually positioned further under the table, finally, the strings protruded 2 cm outside the table edge (*Figure 1B*). In the test, four tables were placed on the arena floor, two artificial flowers were placed 3 cm apart under each table, with one flower connected to a long string (length = 4.5 cm), whereas another flower was attached to a short string (length = 1 cm), and a 3.5 cm string segment (2 cm accessible from outside the table) was positioned with a lateral displacement. The short string and string segment were placed along parallel lines and 1.5 cm apart from each other (*Figure 2*). The position of the string segment was randomly assigned on the left or right side of the short string.

In Experiment 3, a further 18 bumblebees were trained with the same procedure as Experiment 2. In the test, four tables were placed on the arena floor, two artificial flowers were placed 3 cm apart under each table, with one connected to a long string (length = 4.5 cm), a 3.5 cm string segment was placed along with another flower, and a 1 cm gap was created between the string segment and flower (*Figure 2*). If the bees show a preference for the strings connected to flowers, this would indicate that bees with string-pulling experience can recognize connected and disconnected strings.

## Experiments 4-5: Do bumblebees generalize when string colors differ between training and testing?

In Experiment 4, to further verify whether bees can generalize from the string color used during training, bees (n=10) were trained with yellow flowers connected to green strings (length = 4.5 cm), such that strings were visually different from the white strings in the test (*Figure 2*). The training method was the same as Experiment 2 and the test protocol was the same as for Experiment 3. If the bees prefer to pull the connected strings, indicating that bees generalize the color of the strings in the training. In Experiment 5, bumblebees (n=16) were trained with yellow flowers connected to blue strings (length = 4.5 cm; *Figure 2*). The training method was the same as for Experiment 2. The test protocol was the same as for Experiment 3. If the bees prefer to pull the connected strings, this would indicate the bees can generalize from the string color used during training.

## Experiment 6-9: Do bumblebees use image matching to discriminate connected versus disconnected strings?

In Experiment 6, bumblebees (n=15) were trained with a transparent Perspex table in which the edge where the strings protruded was covered with opaque black tape (15×2.5 cm [L×W]; *Figure 1B*). The bees could not see the strings above the table; however, they could receive visual feedback both after the string was pulled out from the table and during the initial stages of training. The test protocol was the same as for Experiment 3. If the bees prefer to pull the connected strings, this would indicate that bees memorize the arrangement of string-connected flowers in this task.

In Experiment 7, bumblebees (n=10) were trained with a piece of black tape covering the front part of the transparent table, and a table completely covered with a green board was placed behind the bees (*Figure 1B*). The distance between the two tables was 2 cm. The entire string was not visible during the initial step of training, and the string disappeared from the bees' view after being pulled out from the table. The test protocol was the same as in Experiment 3. If the bees failed the connectivity task, it would indicate that they used visual cues to solve this task. We trained 22 bees, but 12 of them did not pull the string even once during the test. The bees that did not pull the string were not included in the statistical analysis.

In Experiment 8, bumblebees (n=20) were trained with the same protocol as for Experiment 2. In the test, two artificial flowers were placed 3 cm apart under each table, with one flower connected to a coiled string (length = 20 cm, four turns), this pattern was visually different from the straight string during training. Another flower was attached to a short string (length = 1 cm), a 3.5 cm straight string segment was placed along the short string (*Figure 2*), and a 1 cm gap between the short string and the 3.5 cm string segment was presented. If the bees show a preference for the straight strings (same arrangement in training) over the coiled string or show no preference for either of the two strings, this would indicate that they use image matching to solve this task.

In Experiment 9, bumblebees (n=19) were trained with the same protocol as for Experiment 2. In the test, two artificial flowers were placed under each table, with one flower connected to a coiled string (length = 8 cm, one turn). The other flower was attached to a short string (1 cm), a coiled string segment (length = 6 cm, one turn) was placed along the short string, and there was a 1 cm gap between the flower and the 6 cm coiled string segment (*Figure 2*). The two coiled strings in the test were visually different from a straight string in training. If bumblebees use image matching to discriminate connected and disconnected strings, they will exhibit no preference for either of the two strings.

## Statistical analyses

All statistical analyses were conducted with R version 4.2.0. In each experiment, the percentage of bees pulling connected strings was analyzed with generalized linear mixed models (GLMM) [R Development Core Team, lme4 package]. Colony (>1) was set as random effect. If the bees in an experiment were from a single colony (Experiment 4), the percentage was analyzed with a generalized linear model (GLM). Binomial distribution and logit function were employed for both models. The total number of choices made by each bee was set as weights (*Bates et al., 2015*). Bees' first choices between connected and disconnected strings were analyzed with Chi-square tests. The duration of pulling different kinds of strings and the switching interval were first tested with the Shapiro-Wilk test to assess data normality. The duration data that conform to a normal distribution were compared using linear mixed-effects models (LMM), while the data that deviated from

normality were examined using a generalized linear-mixed model (GLMM). In each model, the duration set as the dependent variable, and string type was considered as a fixed effect, and the bee identity and colony (>1) as random effects. Latency to the first choice was analyzed with GLMM, where the experiment was set as a fixed effect and the colony as a random effect with gamma family. The emmeans package (*Lenth et al., 2024*) was employed to conduct multiple comparisons among different experiments.

## Acknowledgements

We thank Jian Chen (Usda-Ars, Biological Control of Pests Research Unit, Stoneville, Mississippi) for valuable comments and suggestions on the early version of the manuscript. This work was supported by the National Natural Science Foundation of China (32301292) and National Natural Science Foundation of China (32271888). Cwyn Solvi was supported by a Templeton World Charity Foundation project grant (TWCF-2020-0539).

## Additional information

### Funding

| Funder | Grant reference number | Author |
| --- | --- | --- |
| National Natural Science Foundation of China | 32301292 | Chao Wen |
| National Natural Science Foundation of China | 32271888 | Junbao Wen |
| Templeton World Charity Foundation | TWCF-2020-0539 | Cwyn Solvi |

The funders had no role in study design, data collection and interpretation, or the decision to submit the work for publication.

### Author contributions

Chao Wen, Conceptualization, Resources, Data curation, Software, Formal analysis, Supervision, Funding acquisition, Validation, Visualization, Methodology, Writing – original draft, Project administration, Writing – review and editing; Yuyi Lu, Conceptualization, Resources, Data curation, Software, Formal analysis, Validation, Visualization, Methodology, Writing – original draft, Writing – review and editing; Cwyn Solvi, Conceptualization, Resources, Data curation, Formal analysis, Visualization, Methodology, Writing – original draft, Project administration, Writing – review and editing; Shunping Dong, Data curation, Software, Formal analysis, Visualization, Methodology, Writing – review and editing; Cai Wang, Xiujun Wen, Shikui Dong, Conceptualization, Resources, Supervision, Methodology, Writing – review and editing; Haijun Xiao, Conceptualization, Resources, Supervision, Writing – review and editing; Junbao Wen, Conceptualization, Resources, Supervision, Funding acquisition, Methodology, Writing – review and editing; Fei Peng, Conceptualization, Resources, Formal analysis, Supervision, Methodology, Writing – original draft, Writing – review and editing; Lars Chittka, Conceptualization, Resources, Formal analysis, Supervision, Funding acquisition, Validation, Investigation, Methodology, Writing – original draft, Project administration, Writing – review and editing

### Author ORCIDs

Chao Wen  https://orcid.org/0000-0002-5650-3454
Cwyn Solvi  http://orcid.org/0000-0003-2517-6179
Haijun Xiao  https://orcid.org/0000-0002-0832-0493
Fei Peng  https://orcid.org/0000-0002-1637-5611
Lars Chittka  https://orcid.org/0000-0001-8153-1732

Reviewer #1 (Public Review): https://doi.org/10.7554/eLife.97018.3.sa1
Author response https://doi.org/10.7554/eLife.97018.3.sa2

## Additional files

### Supplementary files

• Supplementary file 1. The choices of each bee in different experiments.

• Supplementary file 2. The analysis results of bumblebee's preferences for continuous strings in different string-pulling experiments.

• Supplementary file 3. The analysis results of bumblebee's duration for continuous strings in different string-pulling experiments.

• Supplementary file 4. The analysis results of latency to the first choice in different experiments.

• MDAR checklist

### Data availability

All data generated or analysed during this study are included in the manuscript and supporting files.

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
