## [Editor Report · eLife assessment]

This study provides **valuable** new insights into insect cognition and problem-solving in bumblebees. The authors present **convincing** evidence that bumblebees lack causal understanding in a string-pulling task, and find support for bumblebees instead using image-matching for this task.

---

## [Referee Report · Reviewer #1 (Public Review)]

Summary:

In this paper the researchers aimed to address whether bees causally understand string-pulling through a series of experiments. I first briefly summarize what they did:

- In experiment 1, the researchers trained bees without string and then presented them with flowers in the test phase that either had connected or disconnected strings, to determine what their preference was without any training. Bees did not show any preference.

- In experiment 2, bees were trained to have experience with string and then tested on their choice between connected vs. disconnected string.

- Experiment 3 was similar except that instead of having one option which was an attached string broken in the middle, the string was completely disconnected from the flower.

- In experiment 4, bees were trained on green strings and tested on white strings to determine if they generalize across color.

- In experiment 5, bees were trained on blue strings and tested on white strings.

- In experiment 6, bees were trained where black tape covered the area between the string and the flower (i.e. so they would not be able to see/ learn whether it was connected or disconnected).

- In experiments 2-6, bees chose the connected string in the test phase.

- In experiment 7, bees were trained as in expt 3 and then tested where string was either disconnected or coiled i.e. still being 'functional' but appearing different.

- In experiment 8, bees were trained as before and then tested on string that was in a different coiled orientation, either connected or disconnected.

- In experiments 7 and 8 the bees showed no preference.

Strengths:

I appreciate the amount of work that has gone into these experiments and think they are a nice, thorough set of experiments. I enjoyed reading the paper and felt that it was overall well-written and clear. I think experiment 1 shows that bees do not have an untrained understanding of the function of the string in this context. The rest of the experiments indicate that with training, bees have a preference for unbroken over broken string and likely use visual cues learned during training to make this choice. They also show that as in other contexts, bees readily generalize across different colors.

The 'weaknesses' that I previously listed were dealt with by the authors in the revised version of the manuscript. I think the only point that we disagreed on was relating to the ecological relevance of the task to the bees.

Here is my previous comment:

I think the paper would be made stronger by considering the natural context in which the bee performs this behavior. Bees manipulate flowers in all kinds of contexts, and scrabble with their legs to achieve nectar rewards. Rather than thinking that it is pulling a string, my guess would be that the bee learns that a particular motor pattern within their usual foraging repertoire (scrabbling with legs), leads to a reward. I don't think this makes the behavior any less interesting - in fact, I think considering the behavior through an ecological lens can help make better sense of it.

The authors disagreed, writing the following:

"Here we respectfully disagree. The solving of Rubik s cube by humans could be said to be version of finger movements naturally required to open nuts or remove ticks from fur, but this is somewhat beside the point: it s not the motor

sequences that are of interest, but the cognition involved. A general approach in work on animal intelligence and cognition is to deliberately choose paradigms that are outside the animals daily routines this is what we have done here, in asking whether there is means end comprehension in bee problem solving. Like comparable studies on this question in other animals, the experiments are designed to probe this question, not one of ecological validity."

I think the difference would be that humans know that they are doing a rubik's cube whereas I do not think that the bee knows that it is pulling string- I think the bee thinks that it is foraging on a flower. Therefore, I stand by my statement that I think it's worth considering what the bee is experiencing in this task and how it relates to what it would be doing while foraging. I think that as animal cognition researchers we can design tasks that are distinct from what the animal would naturally encounter to ask specific questions about what they are thinking- but that we can never remove the ecological context since the animal will always be viewing the task through that lens. However, I think this may be a philosophical difference in opinion and I am happy with the manuscript as it stands.

---

## [Author Response]

The following is the authors’ response to the original reviews.

**Public Reviews:**

**Reviewer #1 (Public Review):**
Summary:In this paper, the researchers aimed to address whether bees causally understand string-pulling through a series of experiments. I first briefly summarize what they did:- In experiment 1, the researchers trained bees without string and then presented them with flowers in the test phase that either had connected or disconnected strings, to determine what their preference was without any training. Bees did not show any preference.- In experiment 2, bees were trained to have experience with string and then tested on their choice between connected vs. disconnected string.- experiment 3 was similar except that instead of having one option which was an attached string broken in the middle, the string was completely disconnected from the flower.- In experiment 4, bees were trained on green strings and tested on white strings to determine if they generalize across color.- In experiment 5, bees were trained on blue strings and tested on white strings.- In experiment 6, bees were trained where black tape covered the area between the string and the flower (i.e. so they would not be able to see/ learn whether it was connected or disconnected).- In experiments 2-6, bees chose the connected string in the test phase.- In experiment 7, bees were trained as in experiment 3 and then tested where the string was either disconnected or coiled i.e. still being 'functional' but appearing different.- In experiment 8, bees were trained as before and then tested on a string that was in a different coiled orientation, either connected or disconnected.- In experiments 7 and 8 the bees showed no preference.Strengths:I appreciate the amount of work that has gone into this study and think it contains a nice, thorough set of experiments. I enjoyed reading the paper and felt that overall it was well-written and clear. I think experiment 1 shows that bees do not have an untrained understanding of the function of the string in this context. The rest of the experiments indicate that with training, bees have a preference for unbroken over broken string and likely use visual cues learned during training to make this choice. They also show that as in other contexts, bees readily generalize across different colors.Weaknesses:(1) I think there are 2 key pieces of information that can be taken from the test phase - the bees' first choice and then their behavior across the whole test. I think the first choice is critical in terms of what the bee has learned from the training phase - then their behavior from this point is informed by the feedback they obtain during the test phase. I think both pieces of information are worth considering, but their behavior across the entire test phase is giving different information than their first choice, and this distinction could be made more explicit. In addition, while the bees' first choice is reported, no statistics are presented for their preferences.

We agree with the reviewer that the first choice is critical in terms of what the bumblebees have learned from the training phase. We analyzed the bees’ first choice in Table 1, and we added the tested videos. The entire connected and disconnected strings were glued to the floor, the bees were unable to move either the connected or disconnected strings, and avoid learning behavior during the tests. We added the data of bee's each choice in the Supplementary table.

(2) It seemed to me that the bees might not only be using visual feedback but also motor feedback. This would not explain their behavior in the first test choice, but could explain some of their subsequent behavior. For example, bees might learn during training that there is some friction/weight associated with pulling the string, but in cases where the string is separated from the flower, this would presumably feel different to the bee in terms of the physical feedback it is receiving. I'd be interested to see some of these test videos (perhaps these could be shared as supplementary material, in addition to the training videos already uploaded), to see what the bees' behavior looks like after they attempt to pull a disconnected string.

We added supplementary videos of testing phase. As noted in General Methods, both connected and disconnected strings were glued to the floor to prevent the air flow generated by flying bumblebees’ wings from changing the position of the string during the testing phase. The bees were unable to move either the connected or disconnected strings during the tests, and only attempted to pull them. Therefore, the difference in the friction/weight of pulling the both strings cannot be a factor in the test.

(3) I think the statistics section needs to be made clearer (more in private comments).

We changed the statistical analysis section as suggested by the reviewer.

(4) I think the paper would be made stronger by considering the natural context in which the bee performs this behavior. Bees manipulate flowers in all kinds of contexts and scrabble with their legs to achieve nectar rewards. Rather than thinking that it is pulling a string, my guess would be that the bee learns that a particular motor pattern within their usual foraging repertoire (scrabbling with legs), leads to a reward. I don't think this makes the behavior any less interesting - in fact, I think considering the behavior through an ecological lens can help make better sense of it.

Here we respectfully disagree. The solving of Rubik’s cube by humans could be said to be version of finger-movements naturally required to open nuts or remove ticks from fur, but this is somewhat beside the point: it’s not the motor sequences that are of interest, but the cognition involved. A general approach in work on animal intelligence and cognition is to deliberately choose paradigms that are outside the animals’ daily routines-this is what we have done here, in asking whether there is means-end comprehension in bee problem solving. Like comparable studies on this question in other animals, the experiments are designed to probe this question, not one of ecological validity.

**Reviewer #2 (Public Review):**
Summary:The authors wanted to see if bumblebees could succeed in the string-pulling paradigm with broken strings. They found that bumblebees can learn to pull strings and that they have a preference to pull on intact strings vs broken ones. The authors conclude that bumblebees use image matching to complete the string-pulling task.Strengths:The study has an excellent experimental design and contributes to our understanding of what information bumblebees use to solve a string-pulling task.Weaknesses:Overall, I think the manuscript is good, but it is missing some context. Why do bumblebees rely on image matching rather than causal reasoning? Could it have something to do with their ecology? And how is the task relevant for bumblebees in the wild? Does the test translate to any real-life situations? Is pulling a natural behaviour that bees do? Does image matching have adaptive significance?

We appreciate the valuable comment from the reviewer. Our explanation, which we have now added to the manuscript, is as follows:

“Different flower species offer varying profitability in terms of nectar and pollen to bumblebees; they need to make careful choices and learn to use floral cues to predict rewards (Chittka, 2017). Bumblebees can easily learn visual patterns and shapes of flower (Meyer-Rochow, 2019); they can detect stimuli and discriminate between differently coloured stimuli when presented as briefly as 25 ms (Nityananda et al., 2014). In contrast, causal reasoning involves understanding and responding to causal relationships. Bumblebees might favor, or be limited to, a visual approach, likely due to the efficiency and simplicity of processing visual cues to solve the string-pulling task. ”

As above, it worth noting that our work is not designed as an ecological study, but one about the question of whether causal reasoning can explain how bees solve a string-pulling puzzle. We have a cognitive focus, in line with comparable studies on other animals. We deliberately chose a paradigm that is to some extent outside of the daily challenges of the animal.

**Reviewer #3 (Public Review):**
Summary:This paper presents bees with varying levels of experience with a choice task where bees have to choose to pull either a connected or unconnected string, each attached to a yellow flower containing sugar water. Bees without experience of string pulling did not choose the connected string above chance (experiment 1), but with experience of horizontal string pulling (as in the right-hand panel of Figure 4) bees did choose the connected string above chance (experiments 2-3), even when the string colour changed between training and test (experiments 4-5). Bees that were not provided with perceptual-motor feedback (i.e they could not observe that each pull of the string moved the flower) during training still learned to string pull and then chose the connected string option above chance (experiment 6). Bees with normal experience of string pulling then failed to discriminate between connected and unconnected strings when the strings were coiled or looped, rather than presented straight (experiments 7-8).Weaknesses:The authors have only provided video of some of the conditions where the bees succeeded. In general, I think a video explaining each condition and then showing a clip of a typical performance would make it much easier to follow the study designs for scholars. Videos of the conditions bees failed at would be highly useful in order to compare different hypotheses for how the bees are solving this problem. I also think it is highly important to code the videos for switching behaviours. When solving the connected vs unconnected string tasks, when bees were observed pulling the unconnected string, did they quickly switch to the other string? Or did they continue to pull the wrong string? This would help discriminate the use of perceptual-motor feedback from other hypotheses.

We added the test videos as suggested by the reviewer, and we added the data for each bee's choice. However, both connected and disconnected strings were glued to the floor, and therefore perceptual-motor feedback was equal and irrelevant between the choices during the test.

The experiments are also not described well, for my below comments I have assumed that different groups of bees were tested for experiments 1-8, and that experiment 6 was run as described in line 331, where bees were given string-pulling training without perceptual feedback rather than how it is described in Figure 4B, which describes bees as receiving string pulling training with feedback.

We now added figures of Experiment 6 and 7 in the Figure 1B, and we mentioned that different groups of bees were tested for Experiments 1-9.

The authors suggest the bees' performance is best explained by what they term 'image matching'. However, experiment 6 does not seem to support this without assuming retroactive image matching after the problem is solved. The logic of experiment 6 is described as "This was to ensure that the bees could not see the familiar "lollipop shape" while pulling strings....If the bees prefer to pull the connected strings, this would indicate that bees memorize the arrangement of strings-connected flowers in this task." I disagree with this second sentence, removing perceptual feedback during training would prevent bees memorising the lollipop shape, because, while solving the task, they don't actually see a string connected to a yellow flower, due to the black barrier. At the end of the task, the string is now behind the bee, so unless the bee is turning around and encoding this object retrospectively as the image to match, it seems hard to imagine how the bee learns the lollipop shape.

We agree with the reviewer that while solving the task in the last step during training, the bees don't actually see a string connected to a yellow flower, due to the black barrier. Since the full shape is only visible after the pulling is completed and this requires the bee to “check back” on the entire display after feeding, to basically conclude “ this is the shape that I need to be looking for later”.

Another possibility is that bumblebees might remember the image of the “lollipop shape” while training the bees in the first step, in which the “lollipop shape” was directly presented to the bumblebee in the early step of the training.

We added the experiment suggested by the reviewer, and the result showed that when a green table was placed behind the string to obscure the “lollipop shape” at any point during the training phase, the bees were unable to identify the connected string. The result further supports that bumblebees learn to choose the connected string through image matching.

Despite this, the authors go on to describe image matching as one of their main findings. For this claim, I would suggest the authors run another experiment, identical to experiment 6 but with a black panel behind the bee, such that the string the bee pulls behind itself disappears from view. There is now no image to match at any point from the bee's perspective so it should now fail the connectivity task.Strengths:Despite these issues, this is a fascinating dataset. Experiments 1 and 2 show that the bees are not learning to discriminate between connected and unconnected stimuli rapidly in the first trials of the test. Instead, it is clear that experience in string pulling is needed to discriminate between connected and unconnected strings. What aspect of this experience is important? Experiment 6 suggests it is not image matching (when no image is provided during problem-solving, but only afterward, bees still attend to string connectivity) and casts doubt on perceptual-motor feedback (unless from the bee's perspective, they do actually get feedback that pulling the string moves the flower, video is needed here). Experiments 7 and 8 rule out means-end understanding because if the bees are capable of imagining the effect of their actions on the string and then planning out their actions (as hypotheses such as insight, means-end understanding and string connectivity suggest), they should solve these tasks. If the authors can compare the bees' performance in a more detailed way to other species, and run the experiment suggested, this will be a highly exciting paper

We appreciate the valuable comment from the reviewer. We compared the bees' performance to other species, and conducted the experiment as suggested by the reviewer.

**Recommendations for the authors:**

**Reviewer #1 (Recommendations For The Authors):**
Smaller comments:Line 64: is the word 'simple' needed here? It could also be explained by more complex forms of associative learning, no?

We deleted “simple”.

Methods:Line 230: was it checked that this was high-contrast for the bees?

We added the relevant reference in the revised manuscript.

Line 240: how much sucrose solution was present in the flowers?

We added 25 microliters sucrose solution in the flowers. We added the information in the revised manuscript.

Line 266: check grammar.

We checked the grammar as follows: “During tests, both strings were glued to the floor of the arena to prevent the air flow generated by flying bumblebees’ wings from changing the position of the string.”

Statistical analysis:- What does it mean that "Bees identity and colony were analyzed with likelihood ratio tests"?

Bees identity and colony was set as a random variable. We changed the analysis methods in the revised manuscript, and results of the all the experiments did not changed.

- Line 359: do you mean proportion rather than percentage?

We mean the percentage.

- "the number of total choices as weights" - this should be explained further. This is the number of choices that each bee made? What was the variation and mean of this number? If bees varied a lot in this metric, it might make more sense to analyze their first choice (as I see you've done) and their first 10 choices or something like that - for consistency.

This refers to the total number of choices made by each bumblebee. We added the mean and standard error of each bee’s number of choices in Table 1. Some bees pulled the string fewer than 10 times; we chose to include all choices made by each bee.

- More generally I think the first test is more informative than the subsequent choices, since every choice after their first could be affected by feedback they are getting in that test phase. Or rather, they are telling you different things.

All the bees were tested only once, however, you might be referring to the first choice. We used Chi-square test to analyze the bumblebees’ first choices in the test. It is worth noting that both connected and disconnected strings were glued to the floor. The bees were unable to move either the connected or disconnected strings during the tests, and only attempted to pull them. Therefore,the feedback from pulling either the connected or disconnected strings is the same.

- Line 362: I think I know what you mean, but this should be re-phrased because the "number of" sounds more appropriate for a Poisson distribution. I think what you are testing is whether each individual bee chose the connected or the disconnected string - i.e. a 0 or 1 response for each bee?

We agree with the reviewer that each bee chose the connected or the disconnected string - i.e. a 0 or 1 response for each bee, but not the number. We clarify this as: “The total number of the choices made by each bee was set as weights.”

- Line 364-365: here and elsewhere, every time you mention a model, make it clear what the dependent and independent variables are. i.e. for the mixed model, the 'bee' is the random factor? Or also the colony that the bee came from? Were these nested etc?

We clarify this in the revised manuscript. The bee identity and colony is the random factor in the mixed model.

- Line 368: "Latency to the first choice of each bee was recorded" - why? What were the hypotheses/ predictions here?

The latency to the first choice was intended to see if the bumblebees were familiarizing with the testing pattern. A shorter delay time might indicate that the bumblebees were more familiar with the pattern.

- Line 371: "Multiple comparisons among experiments were.." - do you mean 'within' experiments? It seems that treatments should not be compared between different experiments.

We mean multiple comparisons among different experiments; we clarify this in the revised manuscript.

ResultsExperiment 1: From the methods, it sounded like you both analyzed the bees' first choice and their total no. of choices, but in the results section (and Figure 1) I only see the data for all choices combined here.In table 1 and in the text you report the number of bees that chose each option on their first choice, but there are no statistical results associated with these results. At the very least, a chi square or binomial test could be run.Line 138: "Interestingly, ten out of fifteen bees pulled the connected string in their first choice" - this is presented like it is a significant majority of bees, but a chi-square test of 10 vs 5 has a p-value = 0.1967

We used the Chi square test to analyzed of the bees’ first choice. We also added the analyzed data in the Table 1.

Line 143: "It makes sense because the bees could see the "lollipop shape" once they pulled it out from the table." - this feels more like interpretation (i.e. Discussion) rather than results.

We moved the sentence to the discussion.

Line 162: again this feels more like interpretation/ conjecture than results.

We removed the sentence in the results.

Line 184: check grammar.

We checked the grammar. We changed “task” to “tasks”.

FiguresI really appreciated the overview in Figure 5 - though I think this should be Figure 1? Even if the methods come later in eLife, I think it would be nice to have that cited earlier on (e.g. at the start of the results) to draw the reader's attention to it quickly, since it's so helpful. It also then makes the images at the bottom of what is currently Figure 1 make more sense. I also think that the authors could make it clearer in Figure 5 which strings are connected vs disconnected in the figure (even if it means exaggerating the distance more than it was in real life). I had to zoom in quite a bit to see which were connected vs. not. Alternatively, you could have an arrow to the string with the words "connected" "disconnected" the first time you draw it - and similar labels for the other string conditions.

We appreciate the valuable comment from the reviewer. We changed Figure 5 to Figure 2, and Figure 4 to Figure 1. We cited the Figures at the start of the results. We also changed the gap distance between the disconnected strings. Additionally, we added arrows to indicate “connected” and “disconnected” strings in the Figure.

Figure 1 - I think you could make it clearer that the bars refer to experiments (e.g. have an x-axis with this as a label). Also, check the grammar of the y-axis.

We added the experiments number in the Figures. Additionally, we checked the grammar of the y-axis. We changed “percentages” to “parentage”.

I also think it's really helpful to see the supplementary videos but I think it would be nice to see some examples of the test phase, and not just the training examples.

We added Supplementary videos of the testing phase.

**Reviewer #2 (Recommendations For The Authors):**
Below are also some minor comments:L40: "approaches".

We changed “approach” to “approaches”.

L42: but likely mainly due to sampling bias of mammals and birds.

We changed the sentence as follows: String pulling is one of the most extensively used approaches in comparative psychology to evaluate the understanding of causal relationships (Jacobs & Osvath, 2015), with most research focused on mammals and birds, where a food item is visible to the animal but accessible only by pulling on a string attached to the reward (Taylor, 2010; Range et al., 2012; Jacobs & Osvath, 2015; Wakonig et al., 2021).

L64: remove "in this study"

We removed “in this study”.

L64: simple associative learning of what? Isn't your image matching associative too?

We removed “ simple”.

L97: remove "a" before "connected".

We removed “a” before “connected”.

L136-138: but maybe they could still feel the weight of the flower when pulling?

Because both strings were glued to the floor in the test phase, the feedback was the same and therefore irrelevant. This information is noted in the General Methods.

L161: what are these numbers?

We removed the latency in the revised manuscript.

L167/ Table 1: I realise that the authors never tried slanted strings to check if bumblebees used proximity as a cue. Why?

This was simply because we wanted to focus on whether bumblebees could recognize the connectivity of the string.

Discussion: Why did you only control for colour of the string? What if you had used strings with different textures or smells? Unclear if the authors controlled for "bumblebee smell" on the strings, i.e., after a bee had used the string, was the string replaced by a new one or was the same one used multiple times?

We used different colors to investigate featural generalization of the visual display of the string connected to the flower in this task. We controlled for color because it is a feature that bumblebees can easily distinguish.

Both the flowers and the strings were used only once, to prevent the use of chemosensory cues. We clarify this in the revised manuscript.

L182: since what?

We deleted “since” in the revised manuscript.

L182-188: might be worth mentioning that some crows and parrots known for complex cognition perform poorly on broken strings (e.g., https://doi.org/10.1098/rspb.2012.1998 ; https://doi.org/10.1163/1568539X-00003511 ; https://doi.org/10.1038/s41598-021-94879-x) and Australian magpies use trial and error (https://doi.org/10.1007/s00265-023-03326-6).

We added the following sentences as suggested by the reviewer: “It is worth noting that some crows and parrots known for complex cognition perform poorly on the broken string task without perceptual feedback or learning. For example, New Caledonian crows use perceptual feedback strategies to solve the broken string-pulling task, and no individual showed a significant preference for the connected string when perceptual feedback was restricted (Taylor et al., 2012). Some Australian magpies and African grey parrots can solve the broken string task, but they required a high number of trials, indicating that learning plays a crucial role in solving this task (Molina et al., 2019; Johnsson et al., 2023).”

L193: maybe expand on this to put the task into a natural context?

We added the following sentences as suggested by the reviewer:

“Different flower species offer varying profitability in terms of nectar and pollen to bumblebees; they need to make careful choices and learn to use floral cues to predict rewards (Chittka, 2017). Bumblebees can easily learn visual patterns and shapes of flower (Meyer-Rochow, 2019); they can detect stimuli and discriminate between differently coloured stimuli when presented as briefly as 25 ms (Nityananda et al., 2014). In contrast, causal reasoning involves understanding and responding to causal relationships. Bumblebees might favor, or be limited to, a visual approach, likely due to the efficiency and simplicity of processing visual cues to solve the string-pulling task. ”

L204: is causal understanding the same as means-end understanding?

Means-end understanding is expressed as goal-directed behavior, which involves the deliberate and planned execution of a sequence of steps to achieve a goal. Includes some understanding of the causal relationship (Jacobs & Osvath, 2015; Ortiz et al., 2019). .

L235: this is a very big span of time. Why not control for motivation? Cognitive performance can vary significantly across the day (at least in humans).

Bumblebee motivation is understood to be rather consistent, as those that were trained and tested came to the flight arena of their own volition and were foragers looking to fill their crop load each time to return it to the colony.

L232: what is "(w/w)" ? This occurs throughout the manuscript.

“w/w” represents the weight-to-weight percentage of sugar.

L250: this sentence sounds odd. "containing in the central well.." ?? Perhaps rephrase? Unclear what central well refers to? Did the flowers have multiple wells?

We rephrased the sentence as follows: For each experiment, bumblebees were trained to retrieve a flower with an inverted Eppendorf cap at the center, containing 25 microliters of 50% sucrose solution, from underneath a transparent acrylic table

L268: why euthanise?

The reason for euthanizing the bees is that new foragers will typically only become active after the current ones were removed from the hive.

L270: chemosensory cues answer my concern above. Maybe make it clear earlier.

We moved this sentence earlier in the result.

L273: did different individuals use different pulling strategies? Do you have the data to analyse this? This has been done on birds and would offer a nice comparison.

We analyzed the string-pulling strategies among different individuals, and provided Supplementary Table 1 to display the performances of each individual in different string-pulling experiments.

L365: unclear why both models. Would be nice to see a GLM output table.

The duration of pulling different kinds of strings were first tested with the Shapiro-Wilk test to assess data normality. The duration data that conforms to a normal distribution was compared using linear mixed-effects models (LMM), while the data that deviates from normality were examined with a generalized linear-mixed model (GLMM). We added a GLM and GLMM output table in the revised manuscript.

L377: should be a space between the "." and "This".

We added a space between the “.” and “This”.

L383-390: some commas and semicolons are in the wrong places.

We carefully checked the commas and semicolons in this sentence.

**Reviewer #3 (Recommendations For The Authors):**
Minor commentsLine 32: seems to be missing a word, suggest "the bumblebees' ability to distinguish".

we added “the” in the revised manuscript.

Line 47: it would be good to reference other scholars here, this is the central focus of all work in comparative psychology.

We added the reference in the revised manuscript.

Line 50-61: I think the string-pulling literature could be described in more detail here, with mention of perceptual-motor feedback loops as a competing hypothesis to means-end understanding (see Taylor et al 2010, 2012). It seems a stretch to suggest that "String-pulling studies have directly tested means-end comprehension in various species", when perceptual-motor feedback is a competing hypothesis that we have positive evidence for in several species.

We mentioned the perceptual-motor feedback in the introduction as follow:

“Multiple mechanisms can be involved in the string-pulling task, including the proximity principle, perceptual feedback and means-end understanding (Taylor et al., 2012; Wasserman et al., 2013; Jacobs & Osvath, 2015; Wang et al., 2020). The principle of proximity refers to animals preferring to pull the reward that is closest to them (Jacobs & Osvath, 2015). Taylor et al. (2012) proposed that the success of New Caledonian crows in string-pulling tasks is based on a perceptual-motor feedback loop, where the reward gradually moves closer to the animal as they pull the strings. If the visual signal of the reward approaching is restricted, crows with no prior string-pulling experience are unable to solve the broken string task (Taylor et al., 2012).

However, when a green table was placed behind the string to obscure the “lollipop” structure during the training, the bees could not see the “lollipop” during the initial training stage or after pulling the string from under the table. In this situation, the bees were unable to identify the connected string, further proving that bumblebees chose the connected string based on image matching.

Line 68: suggest remove 'meticulously'.

We removed “meticulously”.

Line 99: This is an exciting finding, can the authors please provide a video of a bee solving this task on its first trial?

We added videos in the supplementary materials.

Line 133: perceptual-motor feedback loops should be introduced in the introduction.

We introduced perceptual-motor feedback loops in the revised manuscript.

Line 136: please clarify the prior experience of these bees, it is not clear from the text.

We clarified the prior experience of these bees as follow: Bumblebees were initially attracted to feed on yellow artificial flowers, and then trained with transparent tables covered by black tape (S7 video) through a four-step process.

Line 138: from the video it is not possible to see the bee's perspective of this occlusion. Do the authors have a video or image showing the feedback the bees received? I think this is highly important if they wish to argue that this condition prevents the use of both image matching and a perceptual-motor feedback loop.

We prevented the use of image matching: the bees were unable to see the flower moving towards them above the table during the training phase in this condition. But the bees may receive visual image both after pulling the string out from the table and in the initial stages of training in this condition.

Line 147: please clarify what experience these bees had before this test.

We added the prior experience of bumblebees before training as follow: We therefore designed further experiments based on Taylor et al. (2012) to test this hypothesis. Bumblebees were first trained to feed on yellow artificial, and then trained with the same procedure as Experiment 2, but the connected strings were coiled in the test.

Line 155: This is a highly similar test to that used in Taylor et al 2012, have the authors seen this study?

We mentioned the reference in the revised manuscript as follows: We therefore designed further experiments based on Taylor et al. (2012) to test this hypothesis.

Line 183: This sentence needs rewriting "Since the vast majority of animals, including dogs 183 (Osthaus et al., 2005), cats (Whitt et al., 2009), western scrub-jays (Hofmann et al.,2016) and azure-winged magpies (Wang et al., 2019) are failing in such tasks spontaneously".

We changed the sentence as suggested by the reviewer as follow: Some animals, including dogs (Osthaus et al., 2005), cats (Whitt et al., 2009), western scrub-jays (Hofmann et al., 2016) and azure-winged magpies (Wang et al., 2019) fail in such task spontaneously.

Line 186: "complete comprehension of the functionality of strings is rare" I am not sure the evidence in the current literature supports any animal showing full understanding, can the authors explain how they reach this conclusion?

We wished to say that few animal species could distinguish between connected and disconnected strings without trial and error learning. We revised the sentence as follows:

It is worth noting that some crows and parrots known for complex cognition perform poorly on broken string task without perceptual feedback or learning. For example, New Caledonian crows use perceptual feedback strategies to solve broken string-pulling task, and no individual showed a significant preference for the connected string when perceptual feedback is restricted (Taylor et al., 2012). Some Australian magpies and African grey parrots can solve the broken string task, but it required a high number of trials, indicating that learning plays a crucial role in solving this task (Molina et al., 2019; Johnsson et al., 2023).

Line 190: the authors need to clarify which part of their study provides positive evidence for this conclusion.

We added the evidence for this conclusion as follows: Our findings suggest that bumblebees with experience of string pulling prefer the connected strings, but they failed to identify the interrupted strings when the string was coiled in the test.

Line 265: was the far end of the string glued only?

The entire string was glued to the floor, not just the far ends of the string.